# Hepatotoxicity of Drugs Used in Multiple Sclerosis, Diagnostic Challenge, and the Role of HLA Genotype Susceptibility

**DOI:** 10.3390/ijms24010852

**Published:** 2023-01-03

**Authors:** Lucy Meunier, Dominique Larrey

**Affiliations:** Liver and Transplantation Unit, Montpellier School of Medicine, 34080 Montpellier, France

**Keywords:** multiple sclerosis, liver injury, drug-induced liver injury, autoimmune hepatitis

## Abstract

Multiple sclerosis (MS) is a chronic demyelinating disease of the central nervous system and the association with other autoimmune diseases is well-documented. There are many therapeutic options for the treatment of MS. Most of the available drugs cause drug-induced liver injury (DILI) to variable extents with heterogeneous clinical and biological manifestations, including liver injury with or without signs of hypersensitivity and autoimmunity. The diagnosis of DILI may be particularly difficult because MS is frequently associated with idiopathic autoimmune hepatitis. Recent advances suggest that MS and immune-mediated DILI could be promoted by genetic factors, including HLA genotype. In addition, some of these drugs may promote hepatitis B virus reactivation. This review explores the potential hepatotoxicity of drugs used to treat MS and the criteria to distinguish DILI from idiopathic autoimmune hepatitis associated with MS. The role of susceptible genes both promoting MS and causing the hepatotoxicity of the drug used for MS treatment is also discussed.

## 1. Introduction

Multiple sclerosis (MS) is a chronic demyelinating disease of the central nervous system characterized by a highly variable and unpredictable course [1]. Generally, the first manifestations of MS occur in young patients and predominantly in females before the age of 40 years [2]. The prevalence is estimated to be around 33 per 100,000, with some variation according to geographic region. Patients with MS have a higher mortality rate than the general population. Despite evidence showing the involvement of acquired, environmental, and genetic factors in MS, the precise events leading to the onset of the disease remain largely unknown [1,2]. Onset is however generally considered to be linked to an autoimmune susceptibility [2,3,4,5]. The association of MS with other autoimmune diseases (celiac disease, myasthenia gravis, rheumatoid arthritis, thyroiditis) is a well-documented phenomenon and could support a role of systemic pathogenesis in the disease [2,3,4,5].

The treatment of MS has markedly changed over the two past decades with the progressive availability of numerous drugs allowing an improvement in the control of disease progression or in the prevention of relapse [6]. Most of these drugs are immunomodulators and their administration appears associated with various effects on the liver, from mild changes in liver enzymes to symptomatic liver injuries with various clinical and biological manifestations [7,8]. Standardized drug-induced liver injury (DILI) criteria have indeed been recently defined by an international group of experts [9,10]. Despite the fact that drugs can induce a range of liver lesion types, acute liver injury is by far the most common clinical event with different presentations. Acute DILI is generally defined on the basis of biochemical blood tests, including serum alanine aminotransferase (ALT), serum aspartate aminotransferase (AST), serum alkaline phosphatase (ALP), serum total and conjugated bilirubin, international normalized ratio (INR), and serum albumin. Case definitions for DILI include one of the following thresholds: (1) increased ALT ≥ 5 times (x) the upper limit of normal (ULN); (2) increased ALP ≥ 2× ULN, particularly if there is a concomitant increase in serum gamma glutamyltransferase (GGT) in the absence of other causes of elevated ALP levels; (3) the combination of increased ALT ≥ 3× ULN and total bilirubin > 2× ULN. In case of abnormal values before starting treatment, a two-fold increase from the baseline value is considered. The type of acute liver injury is defined by the ratio (R) of elevation of baseline ALT to baseline alkaline phosphatase (ALT/ULN)/(ALP/ULN). Acute hepatocellular injury is defined by an increase in ALT > 5× ULN alone or a ratio ALT × ULN/ALP > 5× ULN. Acute cholestatic liver injury is defined by an increase in ALP > 2× ULN alone or with a ratio <2. Acute mixed liver injury is defined by a ratio between 2 and 5.

The assessment of severity is based on the occurrence of jaundice, change in INR > 1.5, lowering of serum albumin, symptoms leading to hospitalization, occurrence of encephalopathy, and multiorgan failure. A case is considered severe when one of these criteria is present [9,10]. Acute DILI may be dose-dependent and easily predictable, like for paracetamol exposure. In contrast, acute DILI may be unpredictable, at therapeutic dose, and in a minority of patients; thus an idiosyncratic phenomenon. This is indeed the most common situation. Idiosyncratic DILI may be associated with signs of hypersensitivity and allergy. Signs of hypersensitivity and allergy include fever, chills, skin rash, arthralgia, and hypereosinophilia. These signs evoke an underlying immunoallergic reaction, and in combination with the occurrence of serum autoantibodies, hypereosinophilia suggests the development of an autoimmune reaction. The diagnosis of DILI is based on chronological and clinical criteria, and the elimination of other common causes, particularly acute viral hepatitis. Different scoring systems have been developed to facilitate the diagnosis, in particular the Roussel Uclaf Causality Assessment Method (RUCAM) and the DILI Network (DILIN) expert method [9,10,11].

The high diversity in liver injuries observed following treatment with drugs for MS is reflected by the heterogeneity of the clinical expressions and mechanisms underlying DILI; idiopathic hepatocellular, cholestatic, or mixed liver injury, with or without signs of hypersensitivity, with or without presence of autoantibodies evoking drug-induced autoimmune hepatitis (AIH) [7,8,12]. Therefore, the distinction between drug-induced autoimmune liver injury (DILI-AIH) and idiopathic autoimmune liver injury may be difficult. Finally, the potential mechanisms of DILI are only partially determined or even unknown.

In this review we explore:(1).The potential hepatotoxicity of drugs used to treat MS and the clinical manifestation of DILI;(2).The criteria to differentiate DILI from idiopathic autoimmunity;(3).Whether DILI is due to an immune-related susceptibility;(4).Whether there are susceptible genes that promote both the development of MS and DILI after treatment of MS.

The review of the cases was based on the international criteria for the definition of DILI and the methods of diagnosis [9,10,11].

## 2. MS Therapies and Drug-Induced Liver Injury

The different disease-modifying therapies (DMTs) and immunomodulating treatments used in the management of MS include interferon beta (IFN-β), glatiramer acetate (GA), natalizumab, fingolimod, mitoxantrone, teriflunomide, dimethyl fumarate, alemtuzumab, daclizumab, and ocrelizumab. Their characteristics and mechanisms of action are shown in Table 1. These drugs are largely available in the USA and Europe, but with some differences in indications.

Antanazzo et al. recently studied DILI secondary to MS treatments. Based on the cases reported by pharmacovigilance (FDA Adverse Event Reporting System) from 2004 to 2016, 11,764 reports of DILI events were extracted. Reports were mainly of female cases (75%) and aged ≥30 years (72%). However, MS occurs predominantly in females and after the age of 30 years. It is therefore impossible to distinguish from this study a higher consumption of drugs from a higher susceptibility to these drugs as there are no data on the comparative population of patients with MS using the same treatments without DILI. In this study, almost all of the patients (99%) had received a monotherapy regimen, and no concomitant therapies were reported in 53% of the cases; 30% of cases required hospitalization. Interferon beta-1a (IFN-β 1a) (n = 3368) and fingolimod (n = 1823) received the highest absolute number of DILI reports, followed by natalizumab (n = 1452). A signal of disproportionate reporting was highlighted for interferon (reporting odds ratio (ROR): 1.94) and mitoxantrone (ROR: 2.52), but also for more recent treatments: alemtuzumab (ROR: 1.59), teriflunomide (ROR: 2.37), and fingolimod (ROR: 2.32). DILI was also separated into "asymptomatic" or "severe liver injury". There were no data on the type of hepatitis or mechanism of toxicity. Several mechanisms were proposed: direct loss/alteration of hepatic cell functions via binding of surface receptors, blockade of liver auto-repair ability, immunosuppressive effect with new infection or reactivation of the previous infection, and immune system imbalance evoking overexpression of specific immunity mediators involved in apoptotic pathways. Overall, this study highlighted that DILI appears to be a common feature shared by a number of MS drugs, including both disease-modifying and symptomatic therapies [7].

### 2.1. Interferon Beta (IFN-β)

IFN-β is a cytokine and belongs to a group of naturally occurring proteins which interact with cell surface receptors to produce immunomodulatory effects. There are two types of IFN-β used in the treatment of MS, IFN-β 1a and IFN-β 1b, which are manufactured under five market forms and administered subcutaneously at different dosages with various periodicities:-Recombinant IFN-β 1b is found under three market forms: Betaseron, Betaferon, and Extavia. Recombinant IFN-β 1b is administered subcutaneously every day at the dose of 250 μg. Extavia is also approved for weekly administration.-Rebif is a recombinant IFN-β 1a that is administered subcutaneously at different doses: 8.8 μg, 22 μg, or 44 μg thrice weekly.-Plegridy is a recombinant peginterferon-β1 that is administered subcutaneously at the doses of 63 μg, 94 μg, or 125 μg every two weeks.

IFN-β is frequently used to prevent relapse of MS. All forms of IFN-β are well known to cause mild liver injury that can occasionally lead to severe liver injury with jaundice. Post-marketing studies suggest that 30–60% of MS patients exposed to an IFN-β have elevations in liver function tests, justifying monitoring of liver function tests during IFN- β treatment. Although fulminant liver failure requiring liver transplantation has been reported in MS patients, only around 1–2% of IFN-β-exposed MS patients experience severe elevations of liver function tests [14,15,16]. The usual delay for elevations in liver function tests is 2–12 months after starting therapy. The clinical pattern is usually hepatocellular. Fever, rash, and eosinophilia are not common. Autoimmune features are somewhat common, but may relate more to the underlying MS rather than DILI. Most reported cases have occurred in females [14,15,16]. The mechanisms of IFN-β hepatotoxicity are only partially documented [14,15,16] with the asymptomatic elevations in transaminases reported as potentially dose dependent. The cases with acute jaundice occasionally associated with autoimmune features may represent the triggering of an underlying autoimmune mechanism [14,15,16].

### 2.2. Methylprednisolone Pulse Therapy

Intravenous (IV) corticosteroid therapy with methylprednisolone (MPDN) is indicated for the treatment of relapse of MS. A recent study by Kimura H et al. investigated the development of liver injury after MPDN pulse therapy. From 2005 to 2016, eight patients (/120, 6.7%) with MS developed liver injury after MPDN pulse therapy. Liver injury usually develops beyond two weeks after MPDN treatment. The clinical profile evoked idiosyncratic acute hepatocellular liver injury; in one case, pathological findings AIH [17]. Assessment of the French pharmacovigilance database allowed the collection of 97 cases of liver injury associated with MPDN from 1985 to 2016 [18]. The prevalence of female sex was 58.8% with a median age of 46 years. MS was the indication for MPDN pulse therapy for 26 cases. MPDN had been administered intravenously in 79.4% of cases. The pattern was predominantly mild/moderate hepatocellular liver injury followed by recovery. A positive rechallenge was observed in 10/13 patients re-exposed to IV administration [18]. Among the cases collected in a Spanish registry, three female cases were reported with acute hepatocellular liver injury at 2–6 weeks after the IV exposure. Rechallenge was positive in two cases [19]. The frequency of liver injury induced by IV MPDN has been addressed by Nociti et al. [20]. They performed a prospective observational study on the risk of liver injury in patients with MS treated with IV MPDN (1000 mg/day for 5 days). The authors collected liver data on a total of 251 cycles of treatment for 175 patients over one year. An increase in transaminases above the ULN (>45 U/L) was observed among 8.6% of patients. The liver injury was significant in 2.5% of the patients, meeting Hy’s law criteria. An extensive diagnostic work-up led to the identification of DILI in three cases and AIH in three other cases. More recently, another prospective study investigating potentially hepatotoxic drugs revealed 13 cases with various neurologic autoimmune diseases occurring with a median latency of 5 weeks after MPDN pulse therapy. Liver injury developed after repeated pulses and was typically hepatocellular with marked severity, leading to liver transplantation in six cases. Histological features showed interface hepatitis and portal fibrosis with mixed inflammatory cells; features suggestive of AIH. Liver injury rapidly responded to prednisone administration [21].

The detailed mechanisms of liver injury caused by MPDN pulse therapy have not been studied. The cases collected in the studies described above show heterogeneity in the clinical and histological findings, sometimes arguing for an idiosyncratic drug reaction, or conversely autoimmune liver injury in some cases. Several hypotheses are proposed as Davidov et al. who propose a direct hepatotoxicity [22], whereas Caçao et al. attribute the development of AIH to MPDN-induced liver injury [23]. In addition, the role of excipients associated with corticosteroids, such as saccharin, can also be evoked as previously described [24,25]. Indeed, whereas MPDN pulse therapy can cause acute liver injury, it is also well-known that oral corticosteroid therapy is currently used to treat liver injuries with autoimmune and allergic components [26,27]. Importantly, liver injury rapidly improved following oral prednisone administration in a German series of cases of DILI associated with MPDN pulse therapy [21].

### 2.3. Glatiramer Acetate (GA) [28,29,30,31,32,33,34,35,36,37,38,39,40,41,42]

GA is a drug made of synthetic polypeptides mimicking myelin proteins. GA acts through an immunomodulating activity of converting pro-inflammatory Th1 cells into regulatory Th2 cells. Thereby, GA decreases inflammation, which leads to a reduced risk of relapse of MS [18].

GA induces the release of cytokines, like IL-4, IL-6, and IL-10, and also enhances the production of autoantibodies. Therefore, one may speculate that GA induces autoimmune side effects. At large, randomized controlled trials of GA in patients with MS, serum ALT elevations were >3× ULN in 7% of GA-treated patients compared to 3% of placebo recipients [8,28]. It is noteworthy that over a dozen cases of clinically apparent liver injury with jaundice have been reported since the approval and the more widespread use of GA [28,29,30,31,32,33,34,35,36,37,38,39,40,41,42].

This low number of DILI cases may indeed explain the absence of a GA-associated risk of DILI in the Antonazzo et al. pharmacovigilance study [7]. The clinical characteristics of the reported cases of GA-associated liver injury are presented in Table 2. Interestingly, six patients that had previously been treated with interferon switched to GA due to abnormal liver assessments, potentially suggesting a predisposition to an autoimmune reaction [31,36,37,38]. The majority of cases involved females with a mean age of 35 years. The delay in symptom onset varied from 7 days to 8 months, and mostly within 1–3 months after starting therapy. The typical presentation was a hepatocellular pattern with serum liver enzyme elevations [28,29,30,31,32,33,34,35,36,37,38,39,40,41,42]. There are arguments in some patients for an autoimmune hepatotoxicity: presence of autoantibodies (including anti-nuclear (A-NA) and anti-smooth muscle (A-SMA) antibodies), liver biopsy showing characteristic histopathological lesions, response to corticosteroid therapy [17]. 

The cases positive for autoantibodies (ANA, ASMA) were not associated with hyperglobulinemia or histologic features of AIH [28,29,30,31,32,33,34,35,36,37,38,39,40,41,42]. After the withdrawal of GA, clinical recovery occurred within 12 weeks [18,19,20,21,22,23,24,25,26,27,28,29,30,31,32]. However, spontaneous autoimmune liver injury occurred in some patients despite the discontinuation of GA [18,19,20,21,22,23,24,25,26,27,28,29,30,31,32]. The mechanism of GA hepatotoxicity remains largely unknown but highlights so far a potential predisposition to autoimmune liver injury. The pattern of GA polypeptide metabolism is more consistent with an autoimmune reaction than a direct hepatotoxicity [18,19,20,21,22,23,24,25,26,27,28,29,30,31,32].

### 2.4. Dimethyl Fumarate (DMF) [33]

DMF is indicated for the substantive treatment of recurrent relapsing forms of MS. DMF reduces inflammation and modulates the activity of the immune system. In an FDA study from 2016, 14 post-marketing cases of clinically-significant DMF-induced liver injury were reported. The liver injury severity was classified as moderate or moderate–severe for eight cases and mild for six. Ten patients required hospitalization but there were no cases leading to liver transplantation or death. Possible mechanisms for DILI associated with DMF include hypersensitivity, AIH, or infection [12,43,44].

### 2.5. Teriflunomide [13,45,46,47,48]

Teriflunomide is an immunomodulatory drug indicated in the prevention of relapse of MS. Teriflunomide is the active metabolite of leflunomide, used to treat rheumatoid poly arthritis, which has shown to cause liver injury with immunoallergic features. Indeed, the effect of teriflunomide on the liver has been carefully evaluated [34]. The oral administration of teriflunomide is frequently associated with an asymptomatic increase in transaminases among 13–15% of patients included in clinical trials. An asymptomatic increase in ALT levels >3× ULN occurred in 6% of teriflunomide-treated patients compared to in 4% of patients receiving placebo [34,35,36,37]. Elevation in transaminases was within the first 6 months of administration. Recovery rapidly followed after withdrawal of teriflunomide, and even among half of the patients with continued administration suggesting an adaptation of the liver to this drug [34,35,36,37]. Very rare cases of symptomatic hepatitis without liver failure have been reported [34,38]. The mechanism of elevation in transaminases is unknown. There is no sign of allergic or autoimmune reaction [34,35,36,37,38].

### 2.6. Alemtuzumab [34,39,40,41]

Alemtuzumab is a recombinant humanized monoclonal antibody against CD52, which is present on B and T cells, monocytes, and natural killer cells. Binding to CD52 produces a depletion of B and T cells. The subsequent cell repopulation is made of B and T cells with a different pattern and a modification in cytokine production towards a less inflammatory profile. Alemtuzumab is indicated in relapsing and remitting MS. Infusion of alemtuzumab is frequently followed by a mild and transient increase of transaminases. A case of liver failure has been observed after several episodes of liver injury with signs of autoimmunity following rechallenge [41]. Alemtuzumab can cause autoimmune diseases. This is likely due to a more rapid CD19+ B cell repopulation in the absence of T cell regulatory mechanisms. An example is Graves’ disease, which develops in approximately 30% of patients up to 3 years after the onset of alemtuzumab treatment [49]. Alemtuzumab can also cause reactivation of hepatitis B virus infection, not only in HBsAg carriers, but also in patients with isolated anti-HBc antibodies with severe liver injury. The reactivation of hepatitis C virus (HCV) has also been observed [34]. In a patient with positive HCV serology, the distinction between HCV reactivation and DILI is provided by the detection of serum HCV RNA by polymerase chain reaction (PCR) and evaluation of alemtuzumab causality. Assessment of the latter is by using the DILIN expert method or the RUCAM.

### 2.7. Natalizumab [13,50,51,52,53]

Natalizumab is a humanized neutralizing IgG4k antibody against α-4 integrin that blocks the migration of leukocytes into the brain. Natalizumab exhibits a potent immunosuppressive activity and is an approved therapy in patients with active relapsing and remitting MS. In phase III studies, the asymptomatic increase of transaminases was similar in the natalizumab-treated patients to the placebo group (5% vs. 3%) [34]. Since marketing of natalizumab, a few cases of liver injury have been reported over a period of more than 10 years [34,42,43,44,45]. In a small series of six cases, liver injury occurred with a very variable delay in onset; after the first administration of natalizumab or much later. Liver injury was associated with the presence of autoantibodies in three patients. Corticosteroid therapy was associated with recovery [44,45].

### 2.8. Ocrelizumab [13,54,55]

Ocrelizumab is a fully humanized monoclonal IgG1 antibody against CD20. Ocrelizumab allows the depletion of pre-B cells, mature B cells, and memory B cells, without the modification of plasma cells or lymphoid stem cells, leading to a reduction in immunogenicity. Ocrelizumab is indicated for the treatment of adult patients with active relapsing forms of MS. Mild-to-moderate serum aminotransferase elevations were reported among 1–2% of patients under ocrelizumab therapy. We are not aware of cases with overt acute liver injury with symptoms or jaundice [13]. Similar to rituximab, ocrelizumab may cause HBV or echovirus reactivation [13,54,55]. In case of acute liver injury, reactivation of HBV may be distinguished from DILI by the detection of HBV DNA in serum in addition to the evaluation of ocrelizumab causality; the latter performed by using the DILIN expert method or the RUCAM.

### 2.9. Cladribine

Cladribine is a synthetic analog of adenosine which induces apoptosis by inhibiting DNA synthesis and repair. Apoptosis is mainly induced in lymphocytes as they are dependent on adenosine deaminase activity, thus decreasing lymphocyte count. In phase III clinical trials, liver enzyme abnormalities were not common and elevated transaminases >5x ULN occurred in less than 2% of patients. The post-marketing experience of cladribine use does not reveal a high risk of hepatotoxicity. However, very rare cases of HBV reactivation have been recorded [56,57,58,59,60].

## 3. Distinction between Drug-Induced Liver Injury (DILI) and Autoimmune Hepatitis (AIH) Associated with MS

AIH is a relatively rare acute or chronic liver disease of unknown etiology characterized by hypergammaglobulinemia, circulating autoantibodies, and liver histopathology showing interface hepatitis [61,62]. AIH is associated with various other diseases considered of autoimmune origin; for instance, Hashimoto’s thyroiditis, Graves’ disease, vitiligo, alopecia, rheumatoid arthritis, diabetes mellitus type-1, inflammatory bowel disease, psoriasis, systemic lupus erythematosus, Sjogren’s syndrome, celiac disease, panniculitis, mononeuritis multiplex, urticaria pigmentosa, idiopathic thrombocytopenic purpura, polymyositis, hemolytic anemia, and uveitis [61]. Concurrence of AIH and MS has rarely been reported, either in untreated patients with MS or after immunomodulatory treatment [63]. In the study by De Seze et al. on patients with MS over 7 years, five untreated patients developed hepatitis [64]. Autoantibodies were absent. Liver histopathology was compatible with AIH (interface hepatitis with piecemeal and centrilobular necrosis) in three out of five patients [64]. Evolution of patients was favorable after the introduction of an immunosuppressive therapy, in particular with azathioprine, corticosteroids, or mycophenolate mofetil [64].

The incidence of AIH varies worldwide depending on the geographic region and the age of onset. In adults, the incidence rate ranges from 0.67/100,000 person-years in Israel to 2/100,000 person-years in New Zealand, and the prevalence ranges from 4/100,000 in Singapore to 42.9/100,000 in Alaskan natives [65]. The prevalence in adults is 31.2/100,000 [66] and in children 3/100,000 [65] in the USA. The prevalence of AIH in a large untreated population of MS patients in Europe was estimated at 0.17%, while the prevalence of AIH in a general population in Europe ranges from 15 to 25 cases per 100,000 inhabitants [61]. This 10× higher prevalence of AIH among patients with MS indicates a potential common pathogenetic mechanism between AIH and MS [64,67]. Additionally, a retrospective analysis of extra-hepatic manifestations revealed a 0.4% prevalence of MS among 562 patients with AIH [67]. This association between MS and AIH raises the question of underlying humoral mechanisms (B cell-mediated immunity) for these two diseases. Several other cases of hepatitis in MS patients have been described since the study by De Seze et al. in 2005 [23]. The majority of patients were under treatment for MS, thus raising the possibility of treatment hepatotoxicity. Many drugs have been associated with the syndrome of drug-induced AIH that shares many features of idiopathic AIH [67]. The distinction between DILI-AIH and AIH may be difficult. The criteria which could help to make this distinction are presented in Table 3. The female sex is more frequently observed in patients with AIH compared to DILI AIH; there is no sex predominance in general for the development of DILI AIH. However, the fact that MS patients are predominantly female means that the number of reported cases of DILI AIH may also be more frequently female, without actually representing a higher treatment susceptibility. Male cases of DILI-AIH should therefore be carefully considered.

There are four categories of criteria for the diagnosis of DILI-AIH: clinical, biochemical, histological, and therapeutic. The clinical criteria include history of liver injury or recent abnormal values in liver enzymes, including transaminases and ALP, before the onset of MS treatment; co-existence of extra-hepatic autoimmune manifestations as indicated above. The absence of pre-existing liver injury and other autoimmune manifestations argue for DILI-AIH. There is no specific biochemical profile of liver injury and various presentation: hepatocellular, cholestatic, or mixed. Serum autoantibodies, e.g., ASMA, ANA, anti-DNA, and anti-liver kidney microsome are equally observed in both idiopathic AIH and DILI-AIH, and their titers do not make a difference. However, the late onset of these antibody productions with regards to the onset of acute liver injury is very uncommon in AIH and more in favor of DILI-AIH. The absence of elevated serum gamma globulin levels or IgG, and a spontaneous improvement of liver abnormalities after withdrawal of the suspected drug, represent strong arguments for DILI. Liver biopsy is generally not necessary to make the diagnosis of DILI-AIH when the acute liver injury exhibits all clinical and biochemical criteria evoking DILI-AIH without signs suggestive of a severe form. For other situations, with mixed criteria and/or signs of a severe form, liver biopsy is indicated to obtain additional hints. The diagnosis of DILI-AIH is supported by the following histological lesions: presence of eosinophil infiltration, absence of interface hepatitis, and absence of fibrosis. In contrast, the association of interface hepatitis and liver fibrosis are strong arguments for AIH. When the distinction between idiopathic AIH and DILI-AIH remains unclear, even after liver biopsy, one may use the response to corticosteroid therapy (which is indicated in these circumstances). Here, the arguments for DILI are a very rapid response to corticosteroid therapy and the absence of relapse of liver injury after a rapid discontinuation of corticosteroid therapy [68,69,70]. Finally, when the diagnosis still remains controversial at this stage, it appears that the analysis of genetic factors, such as HLA genotype, may also be helpful as shown below [71].
ijms-24-00852-t003_Table 3Table 3Criteria that can contribute to the differentiation between drug-induced autoimmune hepatitis and idiopathic autoimmune hepatitis associated with MS [72].
Idiopathic Autoimmune HepatitisDrug-Induced Autoimmune HepatitisClinical presentation

Sex prevalenceFemaleNoneClinical symptomsPredominant AST > ALTNon-specificYesNon-specificNoOther signs of autoimmunityYesNoneImmunology

Autoantibodies

Non-specific (ANA and/or ASMA and/or LKM 1)PresentPresentSpecific (anti-mitochondria 6, LKM 2, CYP1A2)NoYes, for specific drugs:Iproniazid, tienilic acid,dihydralazineIncreased IgG level >1.5x ULNNoYesHistology

PlasmocytesAbsent or rarePresentChronic hepatitis/cirrhosis/fibrosisAbsentFrequently presentInterface hepatitisAbsentFrequentEosinophiliaFrequentAbsentCD4+/CD20+RarePresentHLA genotype

DRB1*1501Increased riskDecreased riskHLA DRB1*03 et *04Not relatedIncreased riskClinical evolution

Response to corticosteroidsRapidVariable, possibly slowerRelapse after stopping corticosteroidsNoneYesImmunosuppressive treatment requiredNoneYes


## 4. The Role of Human Leukocyte Antigen (HLA) in MS Patients Who Develop Hepatitis

DILI is generally modulated by numerous factors categorized in two main groups: environmental/acquired factors and genetic factors. The environmental and acquired factors include sex, age, nutrition status, interaction with other drugs taken simultaneously, the inflammatory pattern of underlying diseases, patient behavior, alcohol intake, etc. Genetic factors are the polymorphisms within various enzymes involved in the metabolism and detoxification of drugs, including cytochrome P-450 superfamily members, N-acetyltransferases, glucuronosyltransferases, glutathione transferases, hepatobiliary transporters, and immunological factors. The role of HLA in the underlying mechanisms of DILI is emerging for an increasing number of drugs as recently reviewed by Fontana et al. [73]. Indeed, a strong association between HLA-I and HLA-II groups and DILI has been evidenced for several drugs and herbal medicines. The main examples are amoxicillin-clavulanic acid and HLA-I A*30:02, HLA-I B*18:01,HLA-II DRB1*15:01/DQB1*06:02; flucoxacillin and HLA-I B*57:01 and B*57:03; minocycline and HLA-I B*35:02; trimethoprim-sulfamethoxazole and HLA-I A*34:02, B*14:01,B*27:02, HLA-B*35:01; isoniazid and HLA-I C*12:02, B*52:01, HLA-II DQA1*03:01; terbinafine and HLA A-I A*33:01; allopurinol and HLA-I A*34:02, B*53:01, B*58:01; green tea and HLA-I B*35:01; lumiracoxib with HLA-II DRB1*15:01/*Polygonum multiflorum* and HLA-I B*35:01 [73]. The association between the susceptibility to MS and HLA has been documented for several decades. More than 20 suceptibility alleles have been identified (references déjà dans la version). The most relevant appears to be HLA-DRB1*15:0 and at a lesser extent HLA-DQB1*06:02,-DQA1*01:02, DRB5*01:01 [74]. Recent data show that the main susceptibility alleles, HLA-DRB1*15:01 and HLA-DRB5*01:01, may have a role in the progression of the disease [75]. Interestingly, there is an interaction between the HLA-DRB1*15:01 allele and the blood level of vitamin D which is lower in patients with MS as compared to a control group. Recent data suggest Vitamin D regulation might have a role in MS [74]. HLA alleles also appear to modulate the response to the drugs used in the treatment of MS. In a recent study performed in Brazil, there was a reduction in the Multiple Disability Status Scale MSSS for patients treated with corticosteroids (DRB1*15:01, DPB1*04:01, DQB1*02:01 and DQB1*03:01), azathioprine (DRB1*03:01, DPB1*04:01, DQB1*03:02, DQB1*06:02, HLA-C*07:02), interferon β-1a 22 mcg (DRB1*11:04, DQB1*03:01 and DQB1*03:02), interferon β-1a 30 mcg (DPB1*02:01, HLA-C*05:01), and interferon β-1b (DQB1*02:01) [76]. One study found a relationship between a better response to interferon β and an increased frequency of HLA-DRB1*04, as well as between the HLA-A*03-B*44-DRB1*04 haplotype and decreased frequency of HLA-B*15 [77]. Another study found an association between the response to glatiramer acetate and HLA-DQB2, as well as other genes [78]. Idiopathic AIH which appears to be relatively frequent in patients with MS as compared to general populations is also associated with carriage of HLA DRB1*03:01/*04:01 alleles. The fact that MS, DILI and AIH are modulated by the susceptibility to various HLA alleles raised the question of potential common links but also supports that HLA typing may serve as a diagnostic tool to differentiate DILI from idiopathic autoimmune in MS patients.

Indeed, the association between HLA and DILI in MS patients could explain the development of an AIH phenotype regardless of treatment. In recent years, several authors have explored hepatitis in MS patients undergoing treatment or not. In treatment-experienced patients, real-world pharmacovigilance findings suggest that DILI could be a common feature of MS drugs [7]. Depending on MS therapy, two types of hepatitis are described, DILI or AIH. The treatments used in MS acts by modifying immunity, which may explain the occurrence of AIH with or without predisposing factors. In cohorts of MS cases with the diagnosis of AIH, 2–9% were considered to be induced by drugs. Conversely, drug-induced AIH accounts for 9% of all DILI [10]. The carriage of HLA DRB1*03:01/*04:01 alleles would promote the diagnosis of idiopathic AIH, while the presence of DILI risk alleles would support the diagnosis of drug-induced AIH. Interestingly, one of the DILI risk alleles, HLA DRB1*15:01, occurs less frequently in association with idiopathic AIH than healthy controls, hence genetic testing aids the decision making in this scenario. According to EASL recommendations, suspected drug-induced AIH among MS patients should be evaluated in detail, including causality assessment, serology, genetic tests, and liver biopsy whenever possible [10]. Data concerning HLA are not always available among the cases found in the literature of hepatitis in MS patients. However, MS has been found associated with a specific allele, HLA-DRB1*15 allele [71,79]. This HLA haplotype is also strongly associated with amoxicillin-clavulanate-induced liver injury, and now recently with some other drugs (notably lumiracoxib) [10]. A common HLA phenotype may promote both MS and immune or autoimmune DILI. This might explain the occurrence DILI in MS and also the AIH phenotype. The sequential occurrence of DILI with different MS drugs in the same patient, e.g., IFN-β and then GA, would be consistent with this view.

## 5. Conclusions

The majority of drugs used for treating MS can cause liver injury and for some of them, the reactivation of hepatitis B virus. DILI under MS therapy frequently exhibits autoimmune features. MS is also frequently associated with AIH, as suggested by the 10-times higher prevalence of AIH in patients with MS. Several hypotheses are proposed to explain this association: a higher susceptibility to idiopathic AIH, as for some other manifestations of autoimmunity, or liver injury induced by MS therapies. Recent data suggest a role of HLA expression. Indeed, a common HLA phenotype for MS and immune or autoimmune DILI may explain the occurrence of DILI in MS as well as the autoimmune phenotype. Future works in this field are required to confirm this hypothesis.

## Figures and Tables

**Table 1 ijms-24-00852-t001:** MS therapies [12,13].

MS Therapies	Mechanism of Action	Dose/Administration
Interferon beta 1a/1b (Avonex^®^, Rebif^®^, Betaferon^®^, Entavia^®^)	The mechanism of action of interferon in MS remains partially unknown: Increased expression of anti-inflammatory cytokines, downregulated expression of pro-inflammatory cytokines	Subcutaneous
Glatiramer acetate (GA) (Copaxone^®^)	GA converts the population of T cells from pro-inflammatory Th1 cells to regulatory Th2 cells that can cross the blood–brain barrier and suppress the inflammatory response	20 mg, 1 inj/day subcutaneous
Natalizumab (Tysabri^®^)	Humanized therapeutic monoclonal antibody blocking α-4 integrin, a component of very late antigen (VLA)-4 on lymphocytes. Inhibition of the interaction between VLA4 and vascular cell adhesion molecule (VCAM) ligand prevents lymphocytes from crossing the blood–brain barrier	150–300 mg/month subcutaneous
Fingolimod (Gilenya^®^)	A sphingosine 1-phosphate analogue that acts as a functional antagonist of sphingosine 1-phosphate receptors. Given lymphocytes are dependent on sphingosine 1-phosphate receptors on their surfaces to egress from the lymphoid tissue, instead they remain trapped resulting in a decrease in the number of circulating lymphocytes	0.5 mg/day oral
Mitoxantrone (Novantrone^®^)	Mitoxantrone is a strong inhibitor of topoisomerase II, an enzyme responsible for the unfolding and repair of damaged DNA	Dose according to weight IV (intravenous)
Teriflunomide (Aubagio^®^)	Inhibits proliferation of autoreactive B and T cells	14mg/day oral
Dimethyl fumarate (Tecfidera^®^)	An antioxidant activity via activation of the transcription factor nuclear-factor-erythroid-2-related factor 2 (Nrf2) and reduction in the release of inflammatory cytokines through inhibition of the transcription factor nuclear-factor κB	240–480 mg/day oral
Alemtuzumab (Lemtrada^®^)	Humanized monoclonal antibody directed against CD52, which is present on the surfaces of lymphocytes and monocytes	IV
Daclizumab (Zinbryta^®^)	Monoclonal antibody acting as an interleukin 2 inhibitor	IV
Ocrelizumab (Ocrevus^®^)	Humanized monoclonal antibody directed against CD20	IV
Cladribine (Mavenclad^®^)	Synthetic analog of adenosine	Oral

**Table 2 ijms-24-00852-t002:** Cases of glatiramer acetete (GA)-associated liver injury.

Case	Sex	Age (Years)	Profile (Hep: Hepatocellular; Cho: Cholestatic)	Antibodies	Liver Biopsy	Previous Treatment before GA	Recovery (Days)	Treatment for Recovery
Deltenre et al. (2009) [29]	F	52	Hep	ANA: 1/320 ASMA: 1/80	Centrilobular damage Lymphocyte Macrophage Eosinophil	MPDN	90	No
Onmez et al. (2013) [30]	F	36	Hep	Negative	Polymorphonuclear-rich mixed-type inflammatory cell reaction	GA + MPDN	36	No
Neuman et al. (2007) [31]	H	71	Hep	ANA: 1/1280	Drug-induced liver-injury without fibrotic changes of the liver	IFN (switch due to elevation in liver function test)	30	Budesonide and mycophenolate mofetil
Antezan et al. (2014) [32]	F	28	Hep	Negative	Hepatocellular necrosis, portal bridging, and portal lymphocytic inflammation		30	No
Subramaniam et al. (2012) [33]	F	31	Hep	ASMA: 1/320	Centrilobular hepatocyte necrosis with portal-venous bridging, along with mild portal and interface hepatitis			
Flaire et al. (2015) [34]	F	56	Hep	Negative	Centrilobular hepatocyte necrosis with inflammatory infiltrates composed of lymphocytes and eosinophils	MPDN	45	No
La Gioia et al. (2014) [35]	F	25	Hep	Negative	Inflammatory infiltration: lymphocytes, histiocytes, plasma cells, and a few eosinophil granulocytes		56	
Makhani et al. (2013) [36]	F	15	Hep	Negative	Lymphocytic inflammatory infiltration with mild portal fibrosis, no plasma cells, and no signs of chronic liver disease	IFN (switch due to elevation in liver function test)	54	No
Fernandez et al. (2015)	F	42	Hep	ANA: 1/640	No biopsy	IFN (switch due to elevation in liver function test)	30	
Sinagra et al. (2013) [38]	F	41	Hep	ANA: 1/320	Moderate interface hepatitis with eosinophilic infiltration and porto-portal fibrosis	IFN (switch due to elevation in liver function test)	30	
Sinagra et al. (2013) [38]	F	29	Hep	ANA: 1/160	Lymphoplasmacytic infiltration with porto-portal fibrosis and slight ductal proliferation	IFN (switch due to elevation in liver function test)		CTC + azathioprine
Almeida et al. (2016) [39]	F	65	Hep	ANA: 1/40 ASMA: 1/40	No biopsy	MPDN	147	
Arruti et al. (2012) [40]	F	46	Hep	Negative	No biopsy			CTC
Von Kalckreuth et al. (2008) [41]	F	42	Cho	ANA and ASMA positive	Severe portal and periportal lymphocytic inflammation with necrosis	IFN (switch due to elevation in liver function test)		CTC + azathioprine
Michels F et al. (2020) [42]	F	23	Hep	Negative	Hepatocyte necrosis CD38-positive lymphocytes			CTC

ANA: anti-nuclear antibodies; CTC: corticosteroids; IFN: Interferon; MPDN: Methylprednisolone; ASMA: anti-smooth muscle antibodies.

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
