# Peer review of "Hepatotoxicity of Drugs Used in Multiple Sclerosis, Diagnostic Challenge, and the Role of HLA Genotype Susceptibility"

_ijms, 2023, doi:10.3390/ijms24010852_

Round 1
Reviewer 1 Report
Dear authors,
It is been a pleasure to see the effort in raising attention about hepatotoxicity in response to drugs that are used for the treatment of multiple sclerosis. Especially the focus on real-world data is important and outstanding. This is an important piece of work and a significant contribution to the field.
Please be aware of this and therefore precise in your statements. You may find my comments and questions.

Author Response
Answers point-by point to the comments of the reviewers
Manuscript ijms-2008471
Hepatotoxicity of drugs used in multiple sclerosis, diagnostic challenge and role of HLA genotype susceptibility
Lucy Meunier1 and Dominique Larrey1
Many thanks for the time and effort from yourselves and the reviewers and the valuable feedback on this manuscript. We have now addressed these comments in full, and point-by-point responses are appended below. In addition, major changes are highlighted in red in the revised version of the manuscript.
We hope that you agree that the manuscript is significantly improved as a result of these changes, and that it is now suitable for publication in International Journal of Molecular Sciences.
Reviewer 1
- Major comments
R1/ It is of utmost importance to raise awareness that treatment regimens may induce liver damage. To classify theliver damage as DILI remains difficult and in the frame of this review questionable as no significant clinical indications and parameters are presented by the authors. The authors may include these or consider rephrasing the title with a broader term like hepatotoxicity and clarify within the article, which alterations in the liver parameters were measuredand whether those would reflect DILI or AIH or general hepatitis.
Some information is needed in the introduction to clarify the term DILI to the reader, i.e. RUCAM score, dose-dependent and independent.
à The title has been modified as recommended. The definition of DILI and classification criteria have been indicated in the introduction lines 43-67.
R2/ It would be appreciated if in table 1 further information could be included:
the prescribed dose and administration, other registered side effects, the references to the stated mechanism ofaction.
à The prescribed dose and the references to the stated mechanism of action have been added
R3/ It would be also interesting to know whether all referenced drugs are approved in the US and Europe and if they are considered similarly for the treatment. Which regimen is the primary treatment, which are used at later stages ofthe disease?
àThese drugs are used in USA and Europe but there are some differences in the indications Differences also exist between USA and Canada. So it makes it rather complicated to precise the indications in each area of the world. Nevertheless we precised the most consensual indications in the text for each drug as recently reviewed by the Swiss Neurologist group.
R4 / From the given information it is difficult to understand, whether the correlation with age in line 62 correlates to stronger onset of the disease or if simply more patients at age > 30 were included in the study. The authors mayplease describe this more in detail.
à The review by Antanazzo et al is describing a population of patients with but does not provide the number of patients exposed to MS drugs without liver injury. Thus, it is not possible to distinguish a higher consumption of drug from a higher susceptibility to these drugs in females. This comment has been added.
R5 / In line 79 the authors state that five forms of IFN-b are present. Do the authors mean the natural isoforms ofthe protein or the drug forms? The authors may also include an explanation on the differences between the natural and the synthetic forms of IFN-b as well as the doses applied.
à All interferon b used in multiple sclerosis are recombinant. This point and the description of the different forms and doses are added in the revised version.
R6/ It would interesting to read if the authors would explain, which parameters were measured to define DILI asasymptomatic or severe under Point 2.
à The severity has been defined according to recent international definition as added in the introduction lines.
R7/ In the paragraph about Methylprednisolone pulse it reads as if this DILI may be dose related, or metabolism related. Are any information’s available that would hint towards a difference from these 8 patients compared to theothers?
The reference used (Kimura H, Takeda A, Kikukawa T, Hasegawa I, Mino T, Uchida- Kobayashi S, et al. Liver injury after methylprednisolone pulse 347 therapy in multiple sclerosis is usually due to idiosyncratic drug-induced toxicity rather than autoimmune hepatitis. Mult Scler 348 Relat Disord. 2020 Jul;42:102065.) states that 8 patients werediagnosed with DILI, but for patient 4 no pathological findings are listed, the authors may take this with care and potentially rephrase it in their own article.
à As suggested, the sentence has been modified as follows:
Liver injury develops usually later than two weeks after methylprednisolone treatment. The clinical profile was an idiosyncratic acute hepatocellular liver injury. In one case, pathological findings suggested an autoimmune hepatitis.
R8/ There are other studies that the authors shall include in their article: doi: 10.1016/j.clinre.2019.12.008 /
DOI: 10.1002/brb3.968
DOI: 10.1177/2050640619840147
doi: 10.1097/MEG.0000000000002334
à As recommended, other studies have been added with corresponding references in the revised manuscript.
R9/ Usually corticosteroids are also used against DILI – the authors may include this consideration in their review.
fphar-2022-820724 1..9 (unibe.ch)
Corticosteroid therapy in drug-induced liver injury: Pros and cons - PubMed (nih.gov)
à We agree with the comment of the reviewer. Consequently, the following sentences have been added in the revised version with the mentioned references:
Indeed, whereas pulse methylpredonisolone can cause acute liver injury, it is well known that oral corticosteroid therapy is currently used to treat liver injury with autoimmune or sometimes allergic components (Einar S. Björnsson et al, Role of Corticosteroids in Drug-Induced Liver Injury. A Systematic Review. Frontiers in Pharmacol 2022;13:1-9; Ping Fang Hu et al Corticosteroid therapy in drug-induced liver injury: Pros and cons. J dig dis 2019;20:122-126)
R10/ Under point 2.4 the authors may also consider to mention the articles:
https://doi.org/10.1007/s40263-021-00842-9
https://doi.org/10.3389/fphar.2019.00837
àReferences have been added.
R11/ It would be appreciated if the authors could explain in point 2.6 what CD52 is and its natural function. Why is it agood target in MS? The authors may further explain Graves disease.
à The effect of CD52 inhibition in MS has been clarified as follows :
Alemtuzumab is a recombinant, humanized monoclonal antibody against CD52, which is present on B and T cells, monocytes, and natural killer cells. Binding to CD52 produces a depletion of B and T cells. The following repopulation is made of B and T cells with different pattern and a modification in cytokine production towards less inflammatory profile.
A comment regarding the occurrence of Graves disease has been added as follows:
Alemtuzumab can cause autoimmune diseases possibly because a more rapid CD19+ B cell repopulation in the absence of T cell regulatory mechanisms. An exemple is Grave disease which occurs in around 30% of patients on to 3 years after the onset of alemtuzumab treatment (Baker D, Herrod SS, Alvarez-Gonzalez C, Giovannoni G, Schmierer K. Interpreting lymphocyte reconstitution data from the pivotal phase 3 trials of alemtuzumab. JAMA Neurol. 2017;74(8):961–9)
R12/ If the reactivation of hepatitis C virus has been shown, how likely is it to be DILI and not hepatitis?
à The following clarification has been added:
In a patient with positive HCV serology, the distinction between HCV reactivation and DILI is provided by the detection of the presence of detectable HCV RNA by PCR in the serum and evaluation of alemtuzumab causality. This assessment is facilitated by using the DILIN expert method or the RUCAM.
R13/ The same for CD20 under point 2.8
à Following the request of the reviewer the mode of action of ocrelizumab has been further detailed and the way to distinguish DILI from viral HBV reactivation has been precised as follows:
Ocrelizumab is a fully humanized monoclonal IgG1 antibody to CD20 which allows the depletion of pre-B cells, mature B cells and memory B cells without modification plasm cells and lymphoid stem cells which leads to a reduction of immunogenicity.
In a patient with positive HBV serology, the distinction between HBV reactivation and DILI is provided by the detection of the presence of detectable HBV DNA by PCR in the serum and evaluation of ocrelizumab causality which can be facilitated by using the DILIN expert method or the RUCAM.
R14/ For point 2.6 and 2.7 the authors mention the increase in transaminase, which other liver parameters weremeasured and which information was drawn from those? ALT, AST, etc.
à In the reports of DILI, the collected data include the classical criteria for the evaluation of the type and severity of liver injury according to international definitions.
The details have been added with details in the introduction as requested by the reviewed
R15/ In none of the cases the authors describe the mentioning of the RUCAM score, which is used to diagnose DILI. Was this not considered in any of the studies?
à In the large majority of the report or series the precise RUCAM score is not indicated.
In some series, a different scoring system has been used. For instance in the series assessing DILI with pulse methylprednisolone.
For these reasons, it appears very complicated to precise the RUCAM score if described for each cases of this review.
R16/ The authors do not explain what HLA mean and how in chapter 3 this relation plays a role. It needs more text to understand what the authors want to state in this chapter. How does the HLA allele association relate to the HLAallele associations of DILI?
à some precisions and references have been added.
R17/ In line 217 the authors relate to European data. The authors may include data from the US
à As requested, the most recent reference of the American guidelines for the management of autoimmune liver injury is now added (Michael L. Volk and Nancy Reau Diagnosis and Management of Autoimmune Hepatitis in Adults and Children: A Patient-Friendly Summary of the 2019 AASLD Guidelines. Clinical Liver disease 2021; 17:85-89).
R18/ More text is needed to understand table 3. The clinical presentation does not seem to be specific enough.What are numbers that would signify the statement for the gender prevalence?
Following the propositions in table 3, the best evaluated criteria seems to be histology, but how often do pathologistand clinicians have access to liver biopsies? How would the authors weigh the criteria themselves? Then the authorsmay add values to underline the statement of table 3.
à More test is now added in the revised version to present the criteria which are used to distinguish drug-induced autoimmune liver injury from idiopathic autoimmune liver injury and to answer to points raised below.
R19/ What are numbers that would signify the statement for the gender prevalence?
à There are unknown since the prevalence should also take into account the number of patients exposed to drugs without liver event.
R20/ Following the propositions in table 3, the best evaluated criteria seems to be histology, but how often dopathologist and clinicians have access to liver biopsies?
à The proportion of cases with or without liver biopsy is impossible to indicate precisely because the indication varies according various criteria (type of clinical center, availability, view/experience of the physician….) . Nevertheless, liver biopsy is much more frequently performed in the most severe cases of acute liver injury, with symptoms, jaundice, biochemical makers of liver deficiency.
This pojnt is added in the revised version.
R21/ How would the authors weigh the criteria themselves? Then the authors may add values to underline thestatement of table 3.
à As recommended , the paragraph regarding the distinction between DILI-AIH and idiopathic autoimmune hepatitis (IAH) has been developed to show the counterpart of each category of arguments, clinical, biological, histological and therapeutic, as well as the place of liver biopsy according to the presentation of liver injury and its severity.
R22/ The authors may be more explicit in the mentioning of risk factors of DILI: non-genetic and genetic Risk factorsfor DILI; doi: 10.1053/j.gastro.2010.04.001; https://doi.org/10.3892/etm.2016.3627
Factors for DILI: doi: 10.1002/hep.23577
à This section has been explicated as requested as follows with the corresponding references.
R23/ Does the argument in line 244 hold true for all medications or only the immunotherapies?
à Can the reviewer clarify the question, we did not understand what exactly is being asked.
- Minor comments
Line 10: with or without Line 41: as
Line 51: point 1 does not read well
Line 89: what does somewhat in numbers mean? Line 153: drug name isspelled with an a
Line 156: order of words does not fit
à Corrections have been done
Reviewer 2 Report
The authors discussed the role of HLA genotype susceptibility in drug-induced liver injury in patients with multiple sclerosis. Some concerns and suggestions are listed as below:
In page 10, what do you mean by saying liver injury without or without hypersensitivity?
In page 16, please provide the full name of DILI.
In page 16, Idiopathic should be idiopathic.
In table 1, the machanism of interferon beta should be revised.
In this manuscript, MS was used, while multiple sclerosis was used in the following text.
The part regarding HLA genotype susceptibility of drug-induced liver injury in patients with multiple sclerosis is too short.
More importantly, the relevant is weak.
Please metion specific susceptible and their roles in drug-induced liver injury in patients with multiple sclerosis. And which kind of treatments?
Author Response
Answers point-by point to the comments of the reviewers
Manuscript ijms-2008471
Hepatotoxicity of drugs used in multiple sclerosis, diagnostic challenge and role of HLA genotype susceptibility
Lucy Meunier1 and Dominique Larrey1
Many thanks for the time and effort from yourselves and the reviewers and the valuable feedback on this manuscript. We have now addressed these comments in full, and point-by-point responses are appended below. In addition, major changes are highlighted in red in the revised version of the manuscript.
We hope that you agree that the manuscript is significantly improved as a result of these changes, and that it is now suitable for publication in International Journal of Molecular Sciences.
Reviewers 2
The authors discussed the role of HLA genotype susceptibility in drug-induced liver injury in patients with multiple sclerosis. Some concerns and suggestions are listed as below:
R1/ In page 10, what do you mean by saying liver injury without or without hypersensitivity?
à Signs of hypersensitivity and allergy include fever, chills, skin rash, arthralgia, hypereosinophilia. This has been added in the revised version.
R2/ In page 16, please provide the full name of DILI.
à Drug-induced liver injury has been added before the abbreviation
R3/ In page 16, Idiopathic should be idiopathic
à Corrections done
R4/ In table 1, the mechanism of interferon beta should be revised.
à Modifications have been added in the manuscript
R5/ In this manuscript, MS was used, while multiple sclerosis was used in the following text.
à Corrections done
R6/ The part regarding HLA genotype susceptibility of drug-induced liver injury in patients with multiple sclerosis is too short.
à As requested by reviewer 1, this section has been completed.
R7/ Please mention specific susceptible and their roles in drug-induced liver injury in patients with multiple sclerosis. And which kind of treatments?
à No specific susceptibility factors are mentioned in the published case series. The evoked mechanism of toxicity is idiosyncratic.
Reviewer 3 Report
Manuscript describes the susceptibility of MS patients to drug-induced liver injury – caused by many of the current therapies available. With diagnosis difficult as the disease itself is associated with hepatitis. The paper reviews the current therapies in relation to DILI – as well as proposing ways to distinguish between DILI and MS related hepatitis.
English language is OK, but with often/consistently small corrections needed throughout the manuscript. With many sentences needing attention in the final edit, especially tenses, missing words, context or abbreviations. One major issue (for me) is that many of the paragraphs feel like they are written as notes and have limited sentence fluidity. Sometimes these short facts are pretty basic and not informative (without further explanation).
There is also a lack of consistency in abbreviations, drug names, liver tox (DILI, hepatotox, liver enzymes, transaminases….) – needs consistency.
Specific comments:
Introduction:
Overall, it is very brief, but probably relevant for the manuscript type.
How were the drugs chosen to include in the review – are these ALL known MS therapies or specifically selected as they have association with DILI? Table 1: not clear where this information came from – need to include relevant references?
Table 1: Interferon in MS remains partially unknown – but what is known (partially)? Add drug target info for those currently missing (or at least keep consistency across all drugs).
Results:
Not clear what ROR means in this context (signal of disproportionate reporting)? Some explanation of how calculated, and what the numbers mean is warranted.
Please include the 5 forms of IFN B mentioned several times.
Text for each drug does seem to be still in “note form” on occasion, and maybe needs editing. For example last paragraph of IFN B section – lots of facts, no detail?
2.3: GA: careful with use of abbreviation and full name – be consistent. Switch from IFNB to GA suggests a predisposition to DILI – not necessarily an immune reaction (although of course is still feasible!). Please make this statement clearer.
Unclear why such large amount of detail for GA – ie Table 2 – and not for the other treatments!! Although I like this table a lot as there is a lot of great and certainly relevant detail included…. Is it possible to include such details for all drugs?
3: MS and AIH. The prevalence of AIH being 10 times higher in MS patients is important – BUT stating that the pathomechanisms are the same is a stretch (98% of MS patients don’t have AIH!). Maybe statement can be weakened/updated.
There are interesting and important facts presented in Table 3 – differentiating between drug and idiopathies hepatitis – that could be explored in more detail in the text. In fact, this is the more important/influential aspect of the review presented.
4: HLA. Stated factors for DILI are a bit weak, and many other aspects missing.
The language in the sections need to be checked. Text is also a little repetitive (to section 3). The overall discussion on HLA and hepatitis susceptibility is minimalistic, should be described in more realistic way, as it isn’t totally convincing.
Minor edits:
- Line 10: ….without or without hyper…
- Line 15: This review proposes to review – odd phrase as the review IS reviewing….
- Line 24 – why add in abbreviated (ie MS) – and then not use it?
- Line 38: DILI isn’t correctly defined
- Line 39: recommend replace biological expression with biological manifestation
- Line 43: replace hepatotoxicity with DILI (occurs throughout the manuscript, need to be consistent)
- Line 46: needs rewording – not clear
- Line 54: DMT needs to be defined
- Line 71: blockade of liver auto-repair ability
- Line 79: …of IFN-b and IFN-b represents…
- Line 84: elevations of liver tests enzymes….justifying monitoring of liver tests enzymes during IFN treatment (also line 86)
- Line 97: A recent study by Kimura H et al, has been interested in reported on the development……
- Line 104: The role of 104 excipients associated with corticosteroids such as saccharin can also be evoked as described previously – reword or add some detail
- Line 111: which leads to reduced the risk of relapse of multiple sclerosis
- Line 30: “Clinical characteristics of reported cases of GA associated liver injury are presented in table 2” and “. Interestingly, 6 patients had previously been treated with interferon switched 131 by GA because of disturbances in the hepatic assessment which may suggest a predisposition to immune reaction (21,26–28). “ are a repeat of same sentences in same paragraph.
- Line 136 and Table 2: ANA, SMA need to be defined
- Line 136: were not associated without – ambiguous (double negative)
- Line 148: …dimethyl fumarate induced liver injury-induced was reported.
- Line 16: consistency - 3 times the upper limit of normal, previously used ULN
- Line 165: ….failure have been reported
- Line 201: not sure it is necessary to list so many other autoimmune diseases.
- Line 206: The cConcurrence
- Line 211: add reference
- Line 213: “The evolution was favorable” – please explain
- Line 228. Sentence doesn’t have an “end” - needs to be competed.
- Lione 270: Most drugs used for treating MS may cause liver injury, and for some of them reactivation of hepatitis B virus – sentence doesn’t make sense?
Author Response
Answers point-by point to the comments of the reviewers
Manuscript ijms-2008471
Hepatotoxicity of drugs used in multiple sclerosis, diagnostic challenge and role of HLA genotype susceptibility
Lucy Meunier1 and Dominique Larrey1
Many thanks for the time and effort from yourselves and the reviewers and the valuable feedback on this manuscript. We have now addressed these comments in full, and point-by-point responses are appended below. In addition, major changes are highlighted in red in the revised version of the manuscript.
We hope that you agree that the manuscript is significantly improved as a result of these changes, and that it is now suitable for publication in International Journal of Molecular Sciences.
Reviewers 3
Manuscript describes the susceptibility of MS patients to drug-induced liver injury – caused by many of the current therapies available. With diagnosis difficult as the disease itself is associated with hepatitis. The paper reviews the current therapies in relation to DILI – as well as proposing ways to distinguish between DILI and MS related hepatitis.
English language is OK, but with often/consistently small corrections needed throughout the manuscript. With many sentences needing attention in the final edit, especially tenses, missing words, context or abbreviations. One major issue (for me) is that many of the paragraphs feel like they are written as notes and have limited sentence fluidity. Sometimes these short facts are pretty basic and not informative (without further explanation).
There is also a lack of consistency in abbreviations, drug names, liver tox (DILI, hepatotox, liver enzymes, transaminases….) – needs consistency.
à The list of abbraviations has been completed. The English was proofread and corrected by a proof editing team.
Specific comments:
Overall, it is very brief, but probably relevant for the manuscript type.
R1/ How were the drugs chosen to include in the review – are these ALL known MS therapies or specifically selected as they have association with DILI? Table 1: not clear where this information came from – need to include relevant references
à References have been added
To review all drugs used in MS, we add cladribine to mention that this drug does not cause significant liver injury as follows: “Cladribine is a synthetic analog of adenosine which induces apoptosis by inhibiting DNA synthesis and repair mainly in lymphocytes which are dependent on adenosine deaminase activity. Cladribine decreases lymophocyte count. In phase 3 clinical trials, abnormalities of liver enzymes were uncommon and transaminase elevation > 5 times the upper limit of normal occurred in less than 2% of patients. Post-marketing experience does not reveals uncommon risk of hepatotoxicity. However very rare cases of HBV reactivation have been recorded. “
R2/ Table 1: Interferon in MS remains partially unknown – but what is known (partially)? Add drug target info for those currently missing (or at least keep consistency across all drugs).
à clarifications have been added
R3/ Not clear what ROR means in this context (signal of disproportionate reporting)? Some explanation of how calculated, and what the numbers mean is warranted.
à ROR means reporting odds ratios, in this study a disproportionality analysis was performed by calculating adjusted reporting odds ratios (RORs). The definition has been added in the manuscript.
R4/ Please include the 5 forms of IFN B mentioned several times.
à The 5 forms of IFN B have been described in the revised version (Refer to answers to Reviewer 1)
R5/ Text for each drug does seem to be still in “note form” on occasion, and maybe needs editing. For example last paragraph of IFN B section – lots of facts, no detail?
à The English was proofread and corrected by a proof editing team.
R6/ 2.3: GA: careful with use of abbreviation and full name – be consistent. Switch from IFNB to GA suggests a predisposition to DILI – not necessarily an immune reaction (although of course is still feasible!). Please make this statement clearer.
à As recommended this point has been clarified as follows: “ Interestingly, 6 patients had previously been treated with interferon switched by GA because of disturbances in the hepatic assessment which suggests a common susceptibility. As these two compounds do not display common chemical structures and are metabolized via different routes, this may suggest a predisposition to an immune reaction which appears more likely than a common metabolic reaction (21,26–28). “
R7/ Unclear why such large amount of detail for GA – ie Table 2 – and not for the other treatments!! Although I like this table a lot as there is a lot of great and certainly relevant detail included…. Is it possible to include such details for all drugs?
à We have chosen to detail only the cases secondary to the GA because they are more frequent and detailed to be focused in a comparative table.
R8/ 3: MS and AIH. The prevalence of AIH being 10 times higher in MS patients is important – BUT stating that the pathomechanisms are the same is a stretch (98% of MS patients don’t have AIH!). Maybe statement can be weakened/updated.
à This sentence has been modified
R9/ There are interesting and important facts presented in Table 3 – differentiating between drug and idiopathies hepatitis – that could be explored in more detail in the text. In fact, this is the more important/influential aspect of the review presented.
à As requested, this part has been extended as presented in the answer to review.
R10/ 4: HLA. Stated factors for DILI are a bit weak, and many other aspects missing.
à This paragraph has been completed as also requested by reviewer 1.
Minor edits: à corrections have been done
- Line 10: ….without or without hyper…
- Line 15: This review proposes to review – odd phrase as the review IS reviewing….
- Line 24 – why add in abbreviated (ie MS) – and then not use it?
- Line 38: DILI isn’t correctly defined
- Line 39: recommend replace biological expression with biological manifestation
- Line 43: replace hepatotoxicity with DILI (occurs throughout the manuscript, need to be consistent)
- Line 46: needs rewording – not clear
- Line 54: DMT needs to be defined
- Line 71: blockade of liver auto-repair ability
- Line 79: …of IFN-b and IFN-b represents…
- Line 84: elevations of liver tests enzymes….justifying monitoring of liver tests enzymes during IFN treatment (also line 86)
- Line 97: A recent study by Kimura H et al, has been interested in reported on the development……
- Line 104: The role of 104 excipients associated with corticosteroids such as saccharin can also be evoked as described previously – reword or add some detail
- Line 111: which leads to reduced the risk of relapse of multiple sclerosis
- Line 30: “Clinical characteristics of reported cases of GA associated liver injury are presented in table 2” and “. Interestingly, 6 patients had previously been treated with interferon switched 131 by GA because of disturbances in the hepatic assessment which may suggest a predisposition to immune reaction (21,26–28). “ are a repeat of same sentences in same paragraph.
- Line 136 and Table 2: ANA, SMA need to be defined
- Line 136: were not associated without – ambiguous (double negative)
- Line 148: …dimethyl fumarate induced liver injury-induced was reported.
- Line 16: consistency - 3 times the upper limit of normal, previously used ULN
- Line 165: ….failure have been reported
- Line 201: not sure it is necessary to list so many other autoimmune diseases.
- Line 206: The cConcurrence
- Line 211: add reference
- Line 213: “The evolution was favorable” – please explain
- Line 228. Sentence doesn’t have an “end” - needs to be competed.
- Lione 270: Most drugs used for treating MS may cause liver injury, and for some of them reactivation of hepatitis B virus – sentence doesn’t make sense?
Round 2
Reviewer 2 Report
In this review, the part regarding HLA genotype susceptibility of drug-induced liver injury in patients with multiple sclerosis is too short. Therefore, the main topic was not discussed in details.
Author Response
The section regarding HLA and MS has been markedly developed.
Reviewer 3 Report
The manuscript is well written, and now much easier to read/digest. All questions/concerns were addressed.
Author Response
We thank the reviewer for the comments and for its contribution to the improvement to the manuscrit.
Round 3
Reviewer 2 Report
no further comments